# COMBINING-ON-GRAPH: STRUCTURED REASONING OF LLMS WITH KNOWLEDGE GRAPH

## ABSTRACT

Large Language Models (LLMs) have demonstrated remarkable performance in reasoning tasks but face challenges such as hallucinations and outdated knowledge, particularly in complex scenarios that require precise and reliable reasoning. Knowledge Graphs (KGs), with their structured, factual nature, provide a promising solution by serving as an external knowledge source to enhance LLMs' performance. However, the vast scale of KGs complicates the retrieval of relevant information. Existing approaches mainly leverage LLMs to *generate* retrieval plans used in logical forms or relation paths for querying KGs, while the generated plans may mismatch with valid relations in KGs, e.g., predict "`language_spoken`" for valid relation "`languages_spoken`". Despite being similar minor inconsistency, it can lead the generated retrieval plan unexecutable. To address this limitation, we propose a novel framework, Combining on Graphs (CoG), where LLMs act as combiners rather than generators. Specifically, rather than directly generating a retrieval plan, CoG encourages LLMs to utilize specified relationships existing within KGs to *combine* a relational path as the retrieval plan. This approach constrains the output space of LLMs to be structured rather than arbitrary, ensuring the generated retrieval plan aligns with the structure of KGs, making it more reliable and adaptable to diverse KGs. Extensive experiments on a range of datasets and reasoning tasks demonstrate that the effectiveness of CoG.

## 1 INTRODUCTION

The development of Large Language Models (LLMs), such as GPT-4 (Achiam et al., 2023), Deepseek-r1 (Guo et al., 2025), and their successors, has brought about significant advancements in natural language processing (NLP). These models, trained on vast amounts of textual data, exhibit exceptional performance across various tasks, including text generation, machine translation, and document understanding (Zhao et al., 2023). By learning from extensive corpora, LLMs have achieved near-human levels of fluency in both language comprehension and generation (OpenAI, 2023; Anthropic, 2024; Guo et al., 2025). Despite their success, in tasks that require precise and reliable reasoning, LLMs still face notable limitations, such as hallucinations (Ji et al., 2023), where the model produces incorrect or fabricated information, and knowledge obsolescence (Lewis et al., 2020), due to the static nature of the training data, which prevents LLMs from incorporating up-to-date or external knowledge.

In response to these challenges, there has been a growing interest in enhancing LLMs with external knowledge sources, particularly Knowledge Graphs (KGs) (Sun et al., 2024; Luo et al., 2024b; Gao et al., 2025). KGs, which encode structured, factual information about entities and their interrelationships, offer a promising solution. By providing a rich, explicitly defined source of verifiable knowledge, KGs can significantly improve LLM performance. However, the integration of KGs with LLMs presents significant challenges. Modern KGs contain vast amounts of information—ranging from general knowledge, such as facts about geographical locations, to highly domain-specific knowledge, such as medical terminology. The sheer scale and complexity of KGs make it difficult to retrieve the most relevant information in a computationally efficient manner.

To address this issue, based on how answers are generated, previous work can be divided into two categories. The first category (Gu & Su, 2022; Luo et al., 2024a) employs a generating-then-retrieving approach, where the question is parsed into a specific logical form (LF) for retrieving

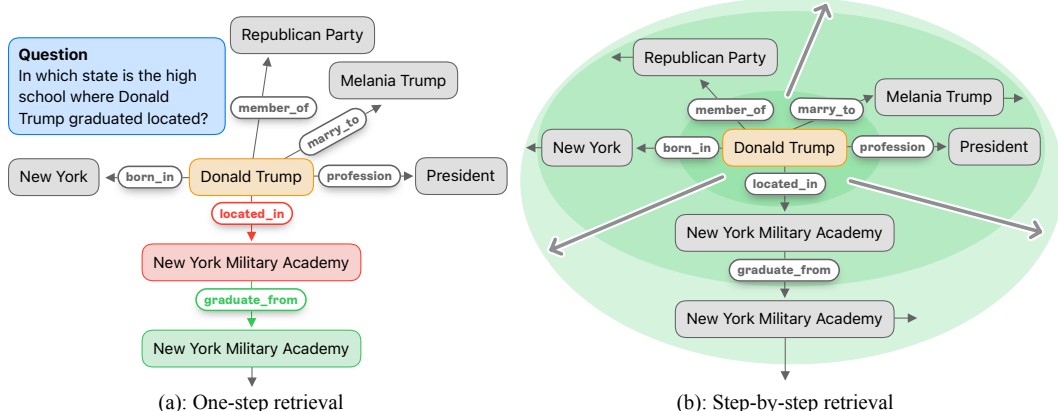

(a): One-step retrieval          (b): Step-by-step retrieval

Figure 1: Step-by-step retrieval vs. One-step retrieval. Step-by-step retrieval performs non-directional, dense searches across all connected relations and entities within KG, resulting in low efficiency. One-step enables directed, sparse searches through predefined retrieval plans.

the answer from KGs. This is known as semantic parsing approach. The second, retrieving-then-generating, involves retrieving relevant knowledge from the KG before generating an answer using LLMs (Sun et al., 2019; Saxena et al., 2022). This is known as retrieval-augmented approach. While semantic parsing-based methods often yield more accurate answers (Das et al., 2021), they are prone to generating non-executable LFs (Yu et al., 2023), which reduces their reliability. Instead, retrieval-augmented methods always generate answers using LLMs, but it still relies on relevant knowledge retrieved from KGs. And recent studies have attempted to leverage additional retrieval LLMs to retrieve knowledge from KGs. These methods typically fall into two categories, as shown in Figure 1. The first, step-by-step retrieval (Sun et al., 2024), involves iteratively expanding the query entity to explore connected relationships and entities. This approach progressively gathers relevant knowledge. The second category, one-step retrieval (Luo et al., 2024b), takes a different approach by pre-generating a retrieval plan, such as a relation path, which is then executed in a single query to retrieve the relevant information more efficiently. While step-by-step retrieval reliably gathers relevant knowledge, it is computationally expensive due to the frequent invocation of LLMs to assess relevance at each step. One-step retrieval, while more efficient, can lead to failures because the retrieval plan generated by the LLM may be arbitrary and fail to conform to the structured relations in the KG. For instance, the model might generate an incorrect or non-existent relationship (e.g., 'languages_spoken' instead of 'language_spoken'), causing the query to fail as the KG does not support such relation.

In this paper, we introduce a novel retrieval-augmented method, Combining on Graphs (CoG), to address key challenges in leveraging LLMs for knowledge graph retrieval. To avoid frequent invocations of LLMs, we aim to leverage LLMs to generate a reliable retrieval plan in advance, so that we can obtain relevant knowledge in a single interaction with KGs. However, LLMs often struggle to generate such retrieval plans that align with the structured schema of KGs, frequently producing hallucinated or invalid relationships that are not grounded in the graph's predefined structure, leading to erroneous or unexecutable outputs. To overcome these limitations, CoG first employs a lightweight model to efficiently explore relevant subgraphs in the KG, identifying key relationships associated with the query. These relationships are incorporated into prompts to guide LLMs, enabling them to comprehend the KG's structural constraints. Specifically, LLMs are encouraged to combine these identified relationships into a coherent and valid retrieval plan, constrained to only use relationships explicitly defined within the KG. This constraint ensures that the generated plan is structurally valid, as it adheres to the KG's schema and prevents the fabrication of non-existent relationships. Finally, the combined retrieval plan is executed within the KGs to retrieve question-relevant knowledge for answering user queries. By anchoring the LLMs' output to the KG's verified structure, CoG significantly mitigates the risks of errors and hallucinations, thereby improving the robustness, reliability, and precision of knowledge extraction for query answering.

Our main contributions can be summarized as follows:

- We propose Combining-on-Graph (CoG), a retrieval-augmented framework that fundamentally shifts how LLMs interact with knowledge graphs—from arbitrary generation to constrained combination of existing KG relations. This addresses a critical failure mode where LLMs generate non-executable retrieval plans due to hallucinated or mismatched relations.

- We introduce a principled approach that restricts LLM outputs to valid KG relations identified through lightweight graph exploration, ensuring structural validity of retrieval plans. Our theoretical analysis shows that this constraint mechanism provably increases the probability of generating correct retrieval paths compared to unconstrained ones (Theorem 3.1).

- Through extensive experiments on WebQSP and CWQ benchmarks, we demonstrate that CoG achieves state-of-the-art performance (86.7% Hits@1 on WebQSP) while maintaining computational efficiency with $5\times$ fewer LLM calls compared to iterative methods. Our ablation studies isolate the impact of each component, validating our design choices.

## 2 RELATED WORK

Currently, KGQA methods primarily consist of two types: retrieval-augmented KGQA and semantic parsing-based KGQA.

**Retrieval-augmented KGQA** extracts relevant information from knowledge graphs to generate answers for complex questions, typically by retrieving subgraphs, entities, or triples pertinent to the query. This process is typically divided into two stages: Information Retrieval and Reasoning. The Information Retrieval stage aims to select question-related knowledge from large-scale knowledge graphs. Dense Passage Retrieval (Karpukhin et al., 2020) implements retrieval using dense representations alone, where embeddings are learned from a small number of questions and passages by a simple dualencoder framework. ToG (Sun et al., 2024) leverages LLMs to assess the relevance between the questions and entities within knowledge graphs as well as relations. In the Reasoning stage, the focus shifts to inferring the final answer based on the retrieved knowledge (Luo et al., 2024b; Mavromatis & Karypis, 2025; He et al., 2021). UniK-QA (Oguz et al., 2020) employs specialized architectures to simulate multi-hop reasoning. However, these approaches often have limited reasoning capabilities. Recently, studies have been focused on leveraging LLMs for reasoning on knowledge graph. ToG (Sun et al., 2024) utilizes the reasoning power of LLMs to iteratively explore multiple reasoning paths within KGs to derive the final answer. RoG (Luo et al., 2024b) introduces a framework for planning retrieval and reasoning by fine-tuning LLMs alongside KGs for more accurate and explainable results. GNN-RAG (Mavromatis & Karypis, 2025) combines language understanding abilities of LLMs with the reasoning abilities of GNNs in a retrieval-augmented generation style. Despite their potential, these approaches frequently face challenges in achieving an optimal trade-off between retrieval efficiency and retrieval accuracy, limiting their performance in complex knowledge graph question-answering scenarios.

**Semantic parsing-based KGQA** converts natural language questions into structured queries, such as SPARQL, for execution on knowledge graphs. These methods aim to map questions to logical forms that capture their semantic structure, enabling precise querying of the knowledge graph. For instance, DECAF (Yu et al., 2023) jointly generates both logical forms and direct answers, and then combines the merits of them to get the final answers. TIARA (Shu et al., 2022) applies multi-grained retrieval to help pretrained language models focus on the most relevant contexts. FM-KBQA (Gao et al., 2025) generates the index of reasoning paths that lead to correct answers by fine-tuning LLMs using multi-task learning, achieving remarkable performance. Despite these advancements, semantic parsing-based methods face challenges, including the need for extensive training data, difficulty in handling colloquial or poorly formed questions, and limited generalizability to open-domain settings where question structures vary widely.

## 3 METHODS

CoG leverages LLMs to formulate robust retrieval strategies, enabling precise knowledge extraction from KGs to address user queries. The methodology systematically integrates a multi-stage process to ensure accurate and efficient reasoning. Specifically, CoG employs LLMs to evaluate the complexity of the query, determining the requisite number of retrieval iterations on KGs. Subsequently, a lightweight model conducts initial exploration to identify pertinent relationships within the graph.

These relationships are then refined and synthesized by LLMs to construct an optimal retrieval plan. This plan facilitates targeted knowledge extraction from the graph, which LLMs subsequently utilizes for rigorous reasoning to derive reliable answers. The process encompasses five distinct stages: Perceiving, Exploring, Planning, Retrieving, and Reasoning.

## 3.1 PERCEIVING AND EXPLORING

CoG generates retrieval plans by integrating relevant relationships within KGs to effectively resolve user queries. To achieve this, an initial exploration of KGs is performed to identify relations pertinent to the query. Given the extensive scale of knowledge graphs, comprehensive exploration is computationally impractical. Prior approaches, such as ToG, required iterative querying of LLMs following each exploration step to evaluate whether the retrieved knowledge sufficed to address the query. This process incurred significant computational costs, leading to inefficient retrieval. In this study, we propose leveraging LLMs' ability to perceive query complexity to determine the necessary number of reasoning steps for resolution. To enable this functionality, we apply Supervised Fine-Tuning (SFT) to the model, formalized as follows:

$$\mathcal{L}_{\text{per}} = -\frac{1}{|\mathcal{D}_{per}|} \sum_{(Q,n)\in\mathcal{D}_{per}} \log P\left(n \mid Q; \phi\right), \tag{1}$$

where Q is the natural language question, n is the predicted number of reasoning steps and $\phi$ . By optimizing the equation 1, we maximize the probability of LLMs perceiving the difficulty of the question.

After perceiving the question, we employ a lightweight model such as BERT (Devlin et al., 2019) to conduct an initial exploration of KGs, thereby filtering out knowledge relevant to the question. To achieve this, we train a classifier using the following objective:

$$\min_{\theta_e} CE\left(\phi_{\theta_e}\left(Q,O\right), Y_e(Q,O)\right), \tag{2}$$

where $O$ is relations in the graph and $Y_e(Q,O) \in \{0,1\}$ stands for the label of the relation $O$. $Y_e(Q,O) = 1$ represents that the relation $O$ is question-relevant, while $Y_e(Q,O) = 0$ means that the relation is irrelevant to the question and will not be included in the context. These selected question-relevant relations $R$ are used to assist in generating retrieval plans.

## 3.2 PLANNING AND RETRIEVING

The key contribution of this paper lies in leveraging LLMs to combine existing relationships within KGs to form retrieval plans, rather than directly using LLMs for generation, thereby avoiding hallucinations. To achieve this, we first conduct preliminary exploration of KGs to filter out relationships relevant to the query. Subsequently, these explored relationships are submitted alongside the user query to the large model to generate reliable retrieval paths:

> Please select appropriate relations from the following set of relations and combine them into a valid relation path that can be helpful for answering the question. Question: {Question}. Relation set: {Relation Set}.

where <question >indicates the question $Q$ and <relations>indicates the question-relevant realations $R$. And such prompt is fed into LLMs to generate the relation path as the retrieval plan: $P = [r_1, r_2, .., r_n]$, where $r_i$ is the $i$-th relation in the retrieval plan. Our learning objective is to maximize the likelihood of the relation $r_i$ given the question Q and the the relations $r_{<i}$, with the following objective function:

$$\mathcal{L}_{\text{plan}} = -\frac{1}{|\mathcal{D}_{plan}|} \sum_{(Q,R,r)\in\mathcal{D}_{plan}} \sum_{i=1}^{|m|} \log p(r_i|Q, R, r_{<i}; \phi), \tag{3}$$

where $R$ is a set of selected candidate relations.

After obtaining the retrieval plan, we can retrieve relevant knowledge $z$ from KGs according to the established plan. Specifically, we start from the question entity $e_0$ on the knowledge graph and traverse the graph along the established retrieval plan until the plan is completed, formulated as:

$$Z = e_0 \xrightarrow{r_1} e_1 \xrightarrow{r_2}, \ldots, \xrightarrow{r_n} e_n. \tag{4}$$

During the retrieval process, there may be multiple valid paths, and we utilize all valid paths for reasoning.

### 3.3 REASONING

After retrieving relevant knowledge from the knowledge graph, we feed it along with the question to LLMs. Leveraging LLMs' reasoning capabilities, it reliably answers user questions based on the retrieved knowledge. The optimization objectives for the reasoning model are as follows:

$$\mathcal{L}_{\text{rea}} = -\frac{1}{|\mathcal{D}_{\text{rea}}|} \sum_{(Q,Z,t) \in \mathcal{D}_{rea}} \sum_{i=1}^{|n|} \log p(r_i | Q, Z, r_{<i}; \phi), \tag{5}$$

where $\log p(r_i | Q, Z, r_{<i}; \phi)$ denotes that LLMs provides the correct answer $r$ based on the retrieved relevant knowledge $Z$, and $r_i$ denotes the tokens of answer $r$.

Finally, the overall loss function for training LLMs is:

$$\mathcal{L} = \mathcal{L}_{\text{per}} + \mathcal{L}_{\text{plan}} + \mathcal{L}_{\text{rea}}, \tag{6}$$

In summary, CoG systematically integrates five stages—Perceiving, Exploring, Planning, Retrieving, and Reasoning—into a robust framework for knowledge extraction and reasoning over KGs. The method leverages the perceptual capabilities of LLMs to assess query complexity, guiding the number of necessary retrieval iterations. A lightweight model, such as BERT, is employed for efficient initial exploration of the KG to filter relevant relations, significantly reducing the computational overhead of exhaustive search methods. During the planning stage, LLMs are utilized to form reliable retrieval paths from the set of relevant relationships, mitigating the risk of hallucinations typically associated with direct generation-based approaches. The refined retrieval plans enable targeted knowledge extraction, which, when fed into LLMs, results in accurate reasoning and reliable answers. This multi-stage approach not only enhances retrieval efficiency but also ensures the robustness and accuracy of the reasoning process, highlighting the potential of combining LLMs with KGs for advanced, scalable KGQA. An overview of our method is presented in Figure 2.

### 3.4 A PROBABILISTIC PERSPECTIVE OF COG

The advantage of CoG over LLMs directly generating retrieval plans from scratch is intuitive: by restricting the generation process to incorporate only legitimate relations as defined in the knowledge graph, CoG eliminates invalid paths, focusing the generation process on viable candidates. To rigorously establish this advantage, we construct a mathematical framework using a Markov chain model, which aligns with the autoregressive nature of LLMs. We compare the traditional unconstrained sequential generation (denoted $P_{\text{gen}}$) with CoG (denoted $P_{\text{com}}$). The following probabilistic framework, through clear definitions and a formal theorem, proves that CoG assigns a higher probability to correct paths, offering a compelling theoretical justification for its superiority.

#### 3.4.1 PRELIMINARIES

We establish the foundational definitions for our sequential generation framework.

**Definition 3.1** (Relation Sets). *Let $\mathcal{R}$ be the finite set of authentic relations in the KG, with cardinality $|\mathcal{R}| = n < \infty$. Let $\mathcal{E}$ be the set of invalid relations not present in the KG. The complete relation vocabulary accessible to the LLM is $\mathcal{V} = \mathcal{R} \cup \mathcal{E}$.*

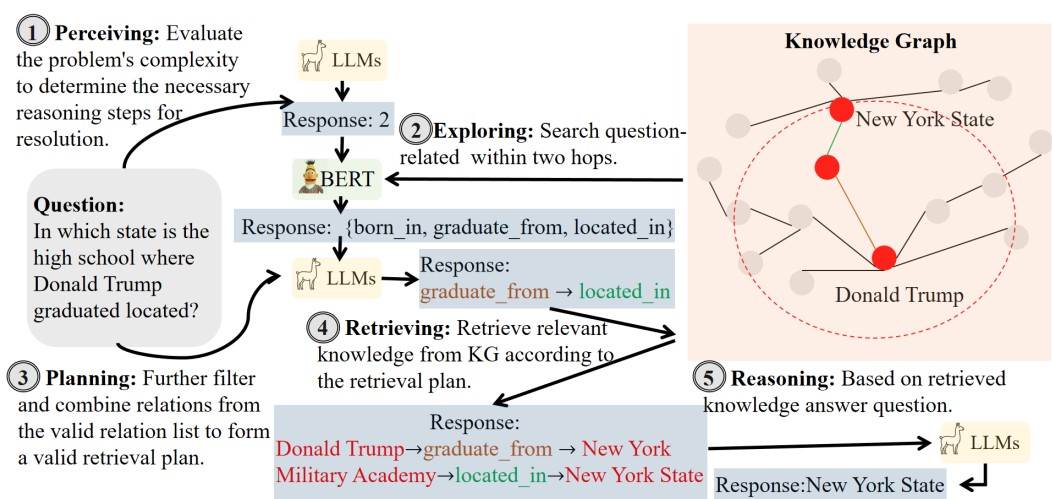

Figure 2: An overview of CoG. CoG operates as a multi-stage pipeline. By systematically combining the retrieved relations from the knowledge graph, it constructs a faithful retrieval plan, enabling effective and structured querying of relevant knowledge.

**Definition 3.2** (Retrieval Paths). *A retrieval path $p = (r_1, r_2, \ldots, r_k)$ is a sequence of relations, where $k \in \mathbb{N}$ is the path length (fixed or variable). Define:*

- *$\Omega$: the set of all possible paths, i.e., the union over possible $k$ of $\mathcal{V}^k$.*

- *$\Omega_{\mathcal{R}} \subset \Omega$: the subset of paths where each relation $r_i \in \mathcal{R}$.*

- *$C \subset \Omega_{\mathcal{R}}$: the set of correct retrieval paths, which are composed of authentic relations ($\mathcal{R}$) and traversable in the KG to reach the target knowledge.*

### 3.4.2 SEQUENTIAL GENERATIVE MODEL

We model the LLM's path generation as a Markov process to reflect its autoregressive behavior.

For the traditional method, path generation follows a Markov chain with transition probabilities $\pi(r_i \mid r_1, \ldots, r_{i-1})$ defined on $\mathcal{V}$, where $r_i$ is the $i$-th relation conditioned on the prefix $(r_1, \ldots, r_{i-1})$. The probability of a path $p = (r_1, \ldots, r_k)$ is:

$$P_{\text{gen}}(p) = \prod_{i=1}^{k} \pi(r_i \mid r_1, \ldots, r_{i-1}).$$

Let $q_i = \sum_{r \in \mathcal{R}} \pi(r \mid r_1, \ldots, r_{i-1})$ denote the probability of selecting an authentic relation at step $i$. We assume $0 < q_i < 1$ for all prefixes, indicating a non-zero probability of generating invalid relations ($\sum_{r \in \mathcal{E}} \pi(r \mid \cdot) > 0$).

Our method, CoG, constrains generation to $\Omega_{\mathcal{R}}$ by using renormalized transition probabilities:

$$\pi_{\mathcal{R}}(r_i \mid r_1, \ldots, r_{i-1}) = \frac{\pi(r_i \mid r_1, \ldots, r_{i-1}) \cdot \mathbf{1}(r_i \in \mathcal{R})}{\sum_{r \in \mathcal{R}} \pi(r \mid r_1, \ldots, r_{i-1})} = \frac{\pi(r_i \mid r_1, \ldots, r_{i-1})}{q_i}, \quad \text{for } r_i \in \mathcal{R},$$

and $\pi_{\mathcal{R}}(r_i \mid \cdot) = 0$ for $r_i \notin \mathcal{R}$, assuming $q_i > 0$. The probability of a path $p \in \Omega_{\mathcal{R}}$ is:

$$P_{\text{com}}(p) = \prod_{i=1}^{k} \pi_{\mathcal{R}}(r_i \mid r_1, \ldots, r_{i-1}).$$

For $p \notin \Omega_{\mathcal{R}}$, $P_{\text{com}}(p) = 0$.

Table 1: Performance comparison with different baselines on the two KGQA datasets.

| Type | Methods | WebQSP | | CWQ | |
|------|---------|--------|---------|-----|---------|
| | | F1 | Hits@1 | F1 | Hits@1 |
| Non-LLMs methods | Rigel | - | 73.3 | - | 48.7 |
| | TIARA | 78.9 | 75.2 | - | - |
| | UniK-QA | 79.1 | - | - | - |
| | UniKGQA | 72.2 | 77.2 | 49.4 | 51.2 |
| Prompting-LLMs Only methods | Zero-shot (gpt-4) | 59.7 | 62.3 | 37.9 | 42.7 |
| | Few-shot (gpt-4) | 62.7 | 68.7 | 43.7 | 51.5 |
| | CoT (gpt-4) | 65.3 | 72.1 | 44.7 | 53.5 |
| Prompting-LLMs + KG | ToG (gpt-3.5) | 72.3 | 75.1 | 56.9 | 57.5 |
| | ToG (gpt-4) | 75.9 | 81.8 | **60.2** | **68.5** |
| | InteractiveKBQA(gpt-4) | 66.3 | 72.9 | 56.5 | 60.3 |
| Finetuning-LLMs + KG | NSM | 62.8 | 68.7 | 42.4 | 47.6 |
| | KD-CoT | 52.5 | 68.6 | 49.7 | 53.3 |
| | DECAF | **78.8** | 82.1 | - | - |
| | RoG | 69.8 | 83.2 | 56.2 | 61.4 |
| | GNN-RAG | 71.3 | 85.7 | 59.4 | 66.8 |
| | CoG | 72.3 | **86.7** | 59.7 | 67.1 |

Within such framework, we have the following theorem:

**Theorem 3.1.** *Assuming $\sum_{r \in \mathcal{E}} \pi(r \mid r_1, \ldots, r_{i-1}) > 0$ (i.e., $q_i < 1$) for all prefixes and $P_{gen}(C) > 0$, CoG yields:*

$$P_{com}(C) > P_{gen}(C).$$

Theorem 3.1 formally establishes that the combination-based retrieval path generation approach employed by CoG yields a higher probability of producing valid paths compared to direct generation methods, thereby providing a robust theoretical foundation for the efficacy of the CoG framework. A detailed proof of this theorem is presented in Appendix A.2.

# 4 EXPERIMENTS

In this section, we first describe the experimental setup in detail, followed by a presentation of the experimental results and a comparison with baseline methods. Subsequently, we conduct ablation studies to evaluate the contribution of individual components of CoG.

## 4.1 SETUP

**Datasets.** Following previous work (Luo et al., 2024b), we evaluate the performance of CoG on two benchmark KGQA datasets: WebQuestionSP (WebQSP) (Yih et al., 2016) and Complex WebQuestions (CWQ) (Talmor & Berant, 2018), with Freebase (Bollacker et al., 2008) as the background knowledge graph. WebQSP contains 4,737 simple natural language questions with SPARQL queries. And CWQ contains 34,689 complex questions with SPARQL queries.

**Implementation details.** For the exploration module, we employ a lightweight model BERT (Devlin et al., 2019) to select question-related relations. This model is trained for 5 epochs on WebQSP and CWQ with a learning rate of 2e-5. For LLMs, Llama-2-7B (Touvron et al., 2023) is fine-tuned and evaluated on both datasets. The model is fine-tuned with a batch size of 16 and a learning rate of 2e-5. And the beam size is set to 5 during beam search in the evaluation process.

**Evaluation metrics.** Following previous works (Luo et al., 2024b; Sun et al., 2024; Gao et al., 2025), we take Hits@1 and F1 score as the evaluation metrics. The Hits@1 represents the overall coverage of the answers and F1 score represents the top-ranked single answer.

Table 2: Faithful Retrieval under CoG.

| Method | Generation | Combination |
|---|---|---|
| **Question** | \multicolumn what religions are popular in france? | |
| **Answer** | Catholicism | |
| **Prompt** | Please generate a valid relation path that can be helpful for answering the following question: {question} | Please select appropriate relations from the following set of relations and combine them into a valid relation path that can be helpful for answering the question. Question: {Question}. Relation set: {Relation Set}. |
| **Retrieval Plan** | location.statistical_region.religion → location.religion_percentage.religion | location.statistical_region.religions → location.religion_percentage.religion |
| **prediction** | None | Catholicism |

Table 3: Efficiency comparison.

| Method | LLM Calls | time (s) | Hits@1 |
|---|---|---|---|
| ToG | 16.3 | 63.9 | 81.8 |
| RoG | 2.0 | 1.5 | 83.2 |
| CoG | 3.0 | 2.3 | 86.7 |

Table 4: CoG with different retriever.

| Retriever | Recall | Hits@1 | F1 |
|---|---|---|---|
| BERT | 89.3 | 86.7 | 72.3 |
| T5 | 88.4 | 86.2 | 71.9 |
| Perfect | 100.0 | 87.4 | 74.6 |

**Baselines.** To ensure a thorough comparison, we compare our method with the following four distinct types of methods: non-LLMs methods, which directly leverages knowledge from KGs to answer user questions; prompting-LLMs only methods, which directly utilizes LLMs to answer user questions; prompting-LLMs + KG methods, which retrieves knowledge from KGs by prompting LLMs to answer user questions; and finetuning-LLMs + KG methods, which combines fine-tuned LLMs with KGs to generate responses. For non-LLMs methods, we include Rigel (Sen et al., 2021), TIARA (Shu et al., 2022), UniK-QA (Oguz et al., 2020) and UniKGQA (Jiang et al., 2023). For prompting-LLMs only methods, We employ three prompting techniques on GPT-4: zero-shot, few-shot (Brown et al., 2020), and CoT prompting (Wei et al., 2022). For prompting-LLMs + KG methods, we include ToG (Sun et al., 2024) and InteractiveKBQA (Xiong et al., 2024). For finetuning-LLMs + KG methods, we include NSM (He et al., 2021), KD-CoT (Wang et al., 2023), DECAF (Yu et al., 2023), RoG (Luo et al., 2024b) and GNN-RAG (Mavromatis & Karypis, 2025).

### 4.2 MAIN RESULTS

**Comparison with other baselines.** As shown in Table 1, we compare our method with other baselines on WebQSP and CWQ. Our approach achieves the best Hits@1 score on WebQSP, outperforming previous state-of-the-art methods. On CWQ, our method demonstrate comparable performance to that achieved using GPT-4. Notably, our approach achieves performance comparable to powerful commercial models using only a fine-tuned Llama2-7b model. This demonstrates the effectiveness of small models on KGQA. Furthermore, our method doesn't require frequent invocations of LLMs to interact with KGs, resulting in higher efficiency. As shown in Table 3, we compare the average number of LLM calls and the time required to answer one question on WebQSP. The results reveal that the one-step retrieval approach of CoG exhibits a significant efficiency advantage over step-by-step retrieval methods like ToG.

**Faithful Retrieval under CoG.** As shown in Figure 2, we present an empirical case study from the CWQ dataset to illustrate the performance of different retrieval plan generation approaches. Both the directly generated retrieval plan and the plan derived from the model's exploration of relational pathways demonstrate a robust understanding of the user query, yielding conceptually sound retrieval strategies. However, the directly generated plan exhibits minor structural inconsistencies with the knowledge graph, rendering it impractical and impeding effective knowledge retrieval. In contrast, the structured reasoning approach employed by CoG ensures that the generated retrieval plans align precisely with the knowledge graph's structure, thereby facilitating successful and efficient knowledge retrieval.

## 4.3 ABLATION STUDY

In this section, we perform ablation experiments. Unless otherwise stated, experiments are conducted on WebQSP.

Table 5: Analysis of the Perceiver.

| Cases | Equal Difficulty | Overestimated Difficulty |
|---|---|---|
| Proportion (%) | 73.86 | 21.21 |

**Analysis of the Perceiver.** CoG utilizes a perceiver-based mechanism to evaluate the complexity of a given question, enabling proactive determination of the optimal number of layers to explore within the knowledge graph for efficient and accurate retrieval. When the perceiver underestimates the complexity of the question, it may result in an incomplete exploration of the knowledge graph, potentially missing critical relations. As presented in Table 5, we evaluate the performance of the perceiver on CWQ by calculating the proportion of queries where its difficulty rating accurately aligned with the actual complexity and where its rating overestimated the actual complexity. In both cases, the knowledge graph was fully traversed. The results demonstrate that the majority of questions can be comprehensively explored within the knowledge graph, ensuring robust retrieval of relevant information.

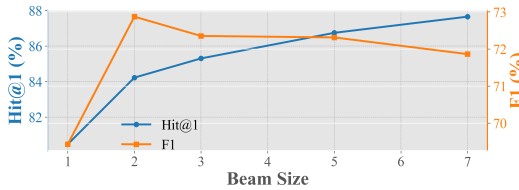

Figure 3: The effect of beam size.

**Analysis of the retriever.** In our method, a small language model is employed for the preliminary exploration of the knowledge graph to identify question-related relations. In the primary experiments, we select BERT as the retriever. Additionally, we evaluate the performance of CoG when using T5 as the retriever and assess its performance under perfect conditions, where all relevant relations are successfully retrieved. As demonstrated in Table 4, even lightweight models such as BERT and T5 achieve high retrieval recall rates. Under ideal conditions, where all relevant relations are retrieved, CoG's performance can be further enhanced.

**The effect of beam size.** As shown in Figure 3, we analyze the effect of beam size on CoG. Beam size is a critical parameter in the beam search decoding strategy employed by large language models to generate responses. It determines the number of candidate sequences retained at each decoding step, balancing exploration of diverse outputs with computational efficiency. To investigate its impact on the performance of CoG, we vary the beam size from 1 to 7 and measure the Hits@1 and F1 score. The results indicate that the F1 stabilizes when the beam size reaches 2, while the Hits@1 increases slightly with increasing beam size. To balance computational cost and performance, we set the beam size to 5 in our experiments.

## 5 CONCLUSION

In this paper, we introduce the Combining on Graphs (CoG) framework, which improves the integration of Knowledge Graphs (KGs) with Large Language Models (LLMs) for reasoning tasks. By enabling LLMs to combine existing relationships within KGs, rather than generate retrieval plans, CoG ensures that the generated plans are structurally aligned with the graph, enhancing reliability and adaptability. Experiments show that CoG outperforms existing methods, addressing issues like hallucinations and outdated knowledge. This work paves the way for more accurate and interpretable knowledge-driven reasoning in LLMs.

## 6 LIMITATIONS

- CoG assumes a consistent and comprehensive KG, but real-world KGs with incomplete or erroneous triples can introduce errors, which may result in a decline in CoG's performance.
- CoG's reliance on fine-tuning Llama 2 to generate accurate retrieval paths introduces significant computational and training overhead, potentially limiting scalability.

## REPRODUCIBILITY STATEMENT

We outline the pipeline of our proposed method in Fig. 2 and provide implementation details in Appendix A.3. The data and code will be released once prepared.

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

# A APPENDIX

## A.1 LLM USAGE STATEMENT

In the preparation of this manuscript, we utilized Large Language Models (LLMs) to assist with language polishing and refinement of the text. Specifically, the LLM was employed to enhance the clarity, coherence, and grammatical accuracy of the writing, ensuring that the manuscript adheres to high standards of academic communication. The LLM did not contribute to the research ideation, methodology, data analysis, or core content development, which were entirely conducted by the authors. All outputs generated by the LLM were carefully reviewed and edited by the authors to ensure alignment with the intended scientific contributions and to maintain the integrity of the work.

## A.2 PROOF

*Proof.* Consider a path $p = (r_1, \ldots, r_k) \in C \subseteq \Omega_{\mathcal{R}}$. The probability under CoG is:

$$P_{\text{com}}(p) = \prod_{i=1}^{k} \pi_{\mathcal{R}}(r_i \mid r_1, \ldots, r_{i-1}) = \prod_{i=1}^{k} \frac{\pi(r_i \mid r_1, \ldots, r_{i-1})}{q_i} = \frac{\prod_{i=1}^{k} \pi(r_i \mid r_1, \ldots, r_{i-1})}{\prod_{i=1}^{k} q_i} = \frac{P_{\text{gen}}(p)}{\prod_{i=1}^{k} q_i}.$$

Since $q_i < 1$, the factor $\frac{1}{\prod_{i=1}^{k} q_i} > 1$.

The probability of generating a correct path is:

$$P_{\text{com}}(C) = \sum_{p \in C} P_{\text{com}}(p) = \sum_{p \in C} \frac{P_{\text{gen}}(p)}{\prod_{i=1}^{k} q_i} = \frac{1}{\prod_{i=1}^{k} q_i} \sum_{p \in C} P_{\text{gen}}(p) = \frac{P_{\text{gen}}(C)}{\prod_{i=1}^{k} q_i}.$$

Since $\prod_{i=1}^{k} q_i < 1$ and $P_{\text{gen}}(C) > 0$, we conclude:

$$P_{\text{com}}(C) > P_{\text{gen}}(C).$$

$\square$

*Remark* A.1. The enhancement factor $\frac{1}{\prod_{i=1}^{k} q_i}$ depends on the path length $k$ and the context-specific $q_i$. This reflects the cumulative effect of constraining each step to authentic relations, amplifying the probability mass on correct paths.

## A.3 IMPLEMENTATION DETAILS

For the exploration module, we employ a lightweight model BERT to select question-related relations. Specifically, we use bert-large-uncased. This model is trained for 5 epochs on WebQSP and CWQ with a learning rate of 2e-5. For LLMs, we use use LLaMA2-Chat-7B. The model is fine-tuned with a batch size of 16 and a learning rate of 2e-5. The training is conducted on 4 H100 GPUs. During inference, the beam size is set to 5 during beam search.

## A.4 DETAILS OF DATASET

In this paper, we conduct experiments on two widely used KGQA datasets: WebQuestionSP (WebQSP) and Complex WebQuestions (CWQ). To ensure a fair comparison, the training and testing splits are the same as in previous works (Luo et al., 2024b; Sun et al., 2018), as shown in Table 6. More detailed statistical results for WebQSP and CWQ are provided in Table 7 8.

Table 6: Split of training and test sets.

| Datasets | #Train | #Test | Max #hop |
|---|---|---|---|
| WebQSP | 2,826 | 1,628 | 2 |
| CWQ | 27,639 | 3,531 | 4 |

Table 7: Statistics of the number of answers to questions in WebQSP and CWQ.

| Dataset | #Ans = 1 | $2 \leq$ #Ans $\leq 4$ | $5 \leq$ #Ans $\leq 9$ | #Ans $\geq 10$ |
|---|---|---|---|---|
| WebQSP | 51.2% | 27.4% | 8.3% | 12.1% |
| CWQ | 70.6% | 19.4% | 6% | 4% |

Table 8: Statistics of the answer hops in WebQSP and CWQ.

| Dataset | 1 hop | 2 hop | $\geq 3$ hop |
|---|---|---|---|
| WebQSP | 65.49% | 34.51% | 0.00% |
| CWQ | 40.91% | 38.34% | 20.75% |

## A.5 DETAILS OF PROMPTS

In the CoG framework, LLMs are strategically deployed to fulfill three primary functions: (1) assess the complexity of input queries; (2) construct retrieval plans; and (3) generate responses based on retrieved knowledge from the knowledge graph.

For the assessment of query complexity, the employed prompt is:

> How many steps do you think are needed to solve the following question: {Question}

In the formulation of retrieval plans, the utilized prompt is:

> Please select appropriate relations from the following set of relations and combine them into a valid relation path that can be helpful for answering the question.
> Question: {Question}.
> Relation set: {Relation Set}.

For the generation of answers, the applied prompt is:

> Based on the reasoning paths, please answer the given question. Please keep the answer as simple as possible and return all the possible answers as a list.
> Reasoning Paths:{Reasoning Paths}.
> Question:{Question}.

This structured utilization of LLMs, guided by tailored prompts, ensures that the retrieval and reasoning processes are aligned with the KG's predefined relational schema, thereby enhancing the accuracy and reliability of the CoG approach.

