# OpenReview forum: "Combining-on-Graph: Structured Reasoning of LLMs with Knowledge Graph"
_ICLR.cc/2026/Conference — Submitted to ICLR 2026_

### Official Review · Reviewer_RPkL · 2025-10-22

**Soundness:** 2
**Presentation:** 2
**Contribution:** 2
**Rating:** 2
**Confidence:** 4

**Summary:**

This paper proposes Combining-on-Graph (CoG), a retrieval-augmented KGQA framework that reframes the role of the LLM from “relation generator” to “relation combiner.” Instead of asking the LLM to generate arbitrary relation paths—which often fail due to hallucinated or invalid relations—CoG first performs lightweight KG exploration (via BERT) to search only valid, question-relevant relations, then constrains the LLM to only combine relations from this vetted set to generate plans. The authors argue that this reduces invalid search space and improves executability. A probabilistic Markov argument formalizes why constraining to valid relations increases the probability of producing a correct path. Experiments on WebQSP and CWQ with fine-tuned LLaMA-2-7B show competitive or state-of-the-art performance with drastically fewer LLM calls.

**Strengths:**

1. It proposed a novel framework that reframes the relation path planning from relation generation to relation combination, which effectively reduces the search space and improves the quality of the generated plans.
2. It proposes a theoretical justification based on a Markov process to explain why constraining the LLM to only combine valid relations increases the probability of producing a correct path.
3. The experimental results on two benchmark datasets (WebQSP and CWQ) demonstrate that CoG achieves competitive or state-of-the-art performance while good efficiency with fewer LLM calls compared to previous methods.

**Weaknesses:**

1. Really constrained generation?: While the paper claims to constrain the LLM to only combine valid relations, it just provides the valid relations as context and relies on the LLM to follow the instruction to generate relation paths. There is no hard guarantee that the LLM will not hallucinate or generate invalid relations outside the provided set, which has been commonly observed in RAG systems. The real constraint, as proposed in GCR [1], actually modifies the decoding process to only allow generating tokens from a constrained vocabulary, avoiding hallucinations.

2. Problematic theoretical analysis: Considering the proposed method is actually a conditioned generation rather than a hard constrained generation, the provided theoretical analysis is problematic. For example, the identity function $1(r_i\in R)$ is not valid in this case since the LLM can still generate relations outside the set $R$.

3. Incomplete baselines: The paper does not compare against several strong STOA baselines in the LLM-augmented KGQA community [1,2,3] . This makes it difficult to fully assess the effectiveness of the proposed approach.

4. No quantitative relation path quality analysis: The paper lacks a detailed analysis of failure cases, which is crucial for comparing the how the proposed method generating valid relation paths compared to other baselines.

5. Generalizability: The proposed method relies on fine-tuning specific BERT models and LLMs for KG exploration, planning and reasoning. It is unclear how well the approach would generalize to other knowledge graphs and QA datasets with unseen relations without fine-tuning.

6. Unclear training details: The paper does not provide sufficient information about the training process, especially how to obtain the question-relevant relation pairs for KG exploration training (Y_e(Q, O)). To the best of my knowledge, this information is not available in the public KGQA datasets used in the experiments.

[1] Luo, Linhao, et al. "Graph-constrained Reasoning: Faithful Reasoning on Knowledge Graphs with Large Language Models." Forty-second International Conference on Machine Learning.

[2] Chen, Liyi, et al. "Plan-on-graph: Self-correcting adaptive planning of large language model on knowledge graphs." Advances in Neural Information Processing Systems 37 (2024): 37665-37691.

[3] Tan, Xingyu, et al. "Paths-over-graph: Knowledge graph empowered large language model reasoning." Proceedings of the ACM on Web Conference 2025. 2025.

**Questions:**

1. Can you provide more details on how the KG exploration model is trained, specifically how the question-relevant relation set is obtained for training (Y_e(Q, O))?

---

### Official Review · Reviewer_b5Rg · 2025-10-31

**Soundness:** 2
**Presentation:** 2
**Contribution:** 1
**Rating:** 2
**Confidence:** 5

**Summary:**

This paper proposes a novel framework, termed combing on graphs (CoG), where LLMs act as combiners rather than generators. To be specific, CoG first leverage a lightweight model to explore relevant subgraphs in the KG to identify key relationships associated with the query. Then, these relationships are incorporated into prompts to guide LLMs to generate valid retrieval plan. Finally, these retrieval plans are used to retrieve question-relevant knowledge for answer generation. Extensive experiments show the effectiveness of the proposed method.

**Strengths:**

1.	This paper is well-organized and easy to follow.
2.	Integrating KGs with LLMs for accurate reasoning is an important research area.
3.	Extensive experiments show the effectiveness of the proposed method.

**Weaknesses:**

1.	There may lack some representative recent baseline methods for comparison, such as GNN-RAG-RA [1], SubgraphRAG [2].
2.	The idea behind the proposed method may lack novelty, as the overall pipeline appears to be borrowed from existing literature. The proposed method may simply add a lightweight retriever to constrain the retrieval-plan generation, and the overall process appears similar to existing retrieve-then-generate methods.
3.	There may be some typos in Section 3.2. In addition, it is unclear what the terms <question> and <relations> refer to, as they have not been mentioned previously.

[1] Mavromatis, Costas, and George Karypis. "Gnn-rag: Graph neural retrieval for large language model reasoning." arXiv preprint arXiv:2405.20139 (2024).

[2] Li, Mufei, Siqi Miao, and Pan Li. "Simple is Effective: The Roles of Graphs and Large Language Models in Knowledge-Graph-Based Retrieval-Augmented Generation." The Thirteenth International Conference on Learning Representations.

**Questions:**

1.	How does the paper ensure that the generated plans are valid, given the stochastic nature of LLM generation even when some constraints are imposed in the prompts?
2.	For the lightweight retriever, is training only the classifier combined with BERT sufficient for subgraph retrieval?
3.	What is the performance when the LLM is directly combined with the retrieved subgraphs for answer generation?

---

### Official Review · Reviewer_Exde · 2025-11-01

**Soundness:** 2
**Presentation:** 2
**Contribution:** 2
**Rating:** 2
**Confidence:** 5

**Summary:**

This paper proposes a framework named Combining-on-Graph (CoG), which shifts the design of retrieval planning from unconstrained relation generation to a structured combination of verified relations in the knowledge graph. This design aims to mitigate execution failures caused by hallucinated or nonexistent relations. By integrating a lightweight relation filtering module and a relation-path combination mechanism, the method improves the executability of retrieval plans while significantly reducing LLM invocation overhead. Experiments conducted on WebQSP and CWQ demonstrate competitive performance, supported by theoretical analysis and ablation studies.

**Strengths:**

**Clear methodological design**: The overall framework is divided into five explicit stages: perception, exploration, planning, retrieval, and reasoning, contributing to a transparent and comprehensible workflow.

**Theoretical justification**: The probabilistic analysis validates that structured relation combination improves the likelihood of generating correct and executable retrieval plans.

**High efficiency**: CoG achieves superior Hits@1 on WebQSP compared to existing methods while significantly reducing the number of LLM calls, demonstrating a desirable trade-off between accuracy and efficiency.

**Weaknesses:**

- **Limited novelty**： The innovation mainly enhances prior frameworks  RoG [1]. Key components (multi-hop prediction and BERT-based relation filtering) have been explored in previous studies[2-3], suggesting an engineering refinement rather than a conceptual breakthrough.

- **Restricted performance gains**: Performance remains behind the state-of-the-art on the more complex CWQ dataset, indicating insufficient capability in handling multi-hop reasoning.

- **Incomplete experimental comparison**: Lacks comparisons against the latest Prompting-LLMs[4-5] and semantic-parsing-based KGQA methods[6], making it difficult to fully validate performance superiority.

- **Insufficient methodological clarity**: Several equations omit key parameter definitions, such as D_per in Eq.(1) and D_plan in Eq.(3), which hinders reproducibility.

- **Inconsistent motivation**: The abstract emphasizes failures caused by logical-form generation, while the Introduction focuses on one-step retrieval errors, leading to a mismatch in motivation descriptions.

- **Unclear visualization**: Figure 1 does not effectively highlight the differences between the two retrieval strategies, and the ordering appears inconsistent with the textual description in the Introduction.

- **Inconsistent related-work categorization**: The classification of KGQA approaches in the Introduction does not align with that in the Related Work section, affecting overall narrative coherence.

  [1] Luo L, Li Y F, Haffari G, et al. Reasoning on Graphs: Faithful and Interpretable Large language Model Reasoning[C]//ICLR 2024: The Twelfth International Conference on Learning Representations. ICLR, 2024.

  [2] Ao T, Yu Y, Wang Y, et al. Lightprof: A lightweight reasoning framework for large language model on knowledge graph[C]//Proceedings of the AAAI Conference on Artificial Intelligence. 2025, 39(22): 23424-23432.

  [3] Sun J, Xu C, Tang L, et al. Think-on-Graph: Deep and Responsible Reasoning of Large Language Model on Knowledge Graph[C]//The Twelfth International Conference on Learning Representations.

  [4] Chen L, Tong P, Jin Z, et al. Plan-on-graph: Self-correcting adaptive planning of large language model on knowledge graphs[J]. Advances in Neural Information Processing Systems, 2024, 37: 37665-37691.

  [5] Tan X, Wang X, Liu Q, et al. Paths-over-graph: Knowledge graph empowered large language model reasoning[C]//Proceedings of the ACM on Web Conference 2025. 2025: 3505-3522.

  [6] Luo H, Haihong E, Tang Z, et al. ChatKBQA: A Generate-then-Retrieve Framework for Knowledge Base Question Answering with Fine-tuned Large Language Models[C]//Findings of the Association for Computational Linguistics ACL 2024. 2024: 2039-2056.

**Questions:**

What are the key innovations of this method compared to RoG? Can BERT-based relation screening and multi-hop prediction be proven necessary and core contributions?

If the Exploration stage results in insufficient relation recall, leading to missing paths, could an error recovery or supplementary mechanism be added?

Although the preliminary experiments mention fine-tuning Llama-2-7B, details and controlled experiments are lacking. For example: Is the gain from fine-tuning quantified and compared? What is the performance difference between the Prompting vs. Fine-tuning modes?

---

### Official Review · Reviewer_z8nu · 2025-11-03

**Soundness:** 3
**Presentation:** 3
**Contribution:** 3
**Rating:** 6
**Confidence:** 3

**Summary:**

The paper introduces **Combining-on-Graph (CoG)**, a KG-augmented LLM framework that replaces free-form, plan-generation with **constrained plan combination**: given a question, a lightweight retriever first selects KG relations likely to be relevant; the LLM then *combines only those valid relations* into a path-style retrieval plan, which is executed on the KG to gather evidence for final answer generation. Training comprises (i) *Perceiving* (predicting reasoning steps), (ii) *Exploring* (BERT/T5 classifier over relations), (iii) *Planning* (next-relation modeling over the candidate set), (iv) *Retrieving* (executing the path), and (v) *Reasoning* (answer generation), with losses (L_{\text{per}}+L_{\text{plan}}+L_{\text{rea}}). A simple Markov-chain analysis (Theorem 3.1) shows that restricting tokens to the valid relation set increases the probability mass on correct paths. Experiments on **WebQSP** and **CWQ** (Freebase) report **86.7% Hits@1** on WebQSP with **fewer LLM calls** than iterative methods.

**Strengths:**

* **Well-motivated constraint:** Switching from unconstrained plan generation to **combination over KG-valid relations** directly targets executability and reduces hallucinated edges.
* **Clean, modular pipeline** with explicit losses (L_{\text{per}},L_{\text{plan}},L_{\text{rea}}) and executable retrieval paths; easy to integrate into existing KGQA stacks.
* **Theory matches intuition:** Theorem 3.1 formalizes the benefit of restricting to valid relations (higher probability of correct paths).
* **Empirical gains with efficiency:** Best WebQSP Hits@1 and fewer LLM calls than iterative ToG/RoG; ablations on retriever choice and beam size provide actionable guidance.

**Weaknesses:**

1. **Narrow evaluation scope.** Only WebQSP/CWQ on Freebase; no Wikidata/DBPedia or text-KG hybrid settings, and no OOD stress tests (edge deletions, alias noise, relation name perturbations).
2. **Limited statistics.** No confidence intervals or multi-seed variance for main tables; efficiency table lacks variability across questions and hardware configs.
3. **Perceiver reliability unclear.** Table 5 suggests frequent overestimation/underestimation is possible, but impact on final accuracy and cost isn’t quantified; the link from predicted steps to exploration depth feels heuristic.
4. **Theory stops short of end metrics.** Theorem 3.1 increases probability of *valid* paths, not necessarily *answer-correct* outcomes under KG incompleteness or ambiguous entity linking. A discussion or bound connecting to Hits@1/F1 would improve the story.
5. **Assumptions and engineering details.** Relation-retriever labels, negative sampling, and candidate-set sizes are not fully specified, making reproduction of the explorer less certain (appendix helps but remains high-level).

**Questions:**

1. **Perceiver supervision.** How are ground-truth step counts annotated for (L_{\text{per}})? If derived from SPARQL path length, how do you handle branching/union queries? Please report perceiver accuracy by hop bin and the downstream effect on Hits@1/F1.
2. **Relation-retriever labeling.** How are positives/negatives built for the relation classifier (Y_e(Q,O))? What is the average candidate-set size (|R|) per question, and how does performance vary with recall@k of the explorer?
3. **Robustness.** Can you report stress tests: (i) drop a fraction of gold edges, (ii) perturb relation names (synonyms/plurals), (iii) add spurious edges—then measure plan executability and answer accuracy?
4. **Statistics and compute.** Please add **means ± 95% CI over ≥3 seeds** for Tables 1/3 and training/inference compute (GPU hours, tokens).
5. **Generality beyond Freebase.** Any results on Wikidata or a text-augmented regime (e.g., Freebase+Wikipedia) to demonstrate portability?
6. **Failure analysis.** In Table 2’s example, what proportion of generation-only plans fail due to **non-existent** relations vs. **type/arg constraints**? Can CoG incorporate type constraints during combination?

---

### Meta-Review · Area_Chair_YDmK · 2026-01-07

**Summary:**

Three reviewers raised consistent concerns: limited evaluation, insufficient comparison with recent strong baselines, questions about technical novelty, lack of statistical robustness reporting, and ambiguous training details. No author rebuttal was submitted, so these concerns remain unaddressed.

**Reviewer Concerns:**

Since the authors did not provide a rebuttal, none of the reviewers’ concerns were addressed during the discussion phase.

**Reviewer Scores:**

No rebuttal was submitted, and no reviewer had the opportunity to revise their assessment based on author responses. Therefore, all scores are unchanged.

---

### Decision · Program_Chairs · 2026-01-26

Reject